# InVEST Model-Based Spatiotemporal Analysis of Water Supply Services in the Zhangcheng District

**Run Liu** [1,2,3], **Xiang Niu** [1,2,3,*], **Bing Wang** [1,2,3] **and Qingfeng Song** [1,2,3]

[1] Research Institute of Forest Ecology, Environment and Protection, Chinese Academy of Forestry, Beijing 100091, China; liurun6046@163.com (R.L.); wangbing@caf.ac.cn (B.W.); songqingfeng@caf.ac.cn (Q.S.)

[2] Key Laboratory of Forest Ecology and Environment, State Forestry and Grassland Administration, Beijing 100091, China

[3] Dagangshan National Key Field Observation and Research Station for Forest Ecosystem, Xinyu 338033, China

[*] Correspondence: niuxiang@caf.ac.cn; Tel.: +86-10-6288-9334

**Abstract:** The Zhangcheng District is critically responsible for protecting water resources, preserving sand sources, and improving the ecological environment in Beijing. Quantitative evaluation and research on the ecosystem water supply services in this area are beneficial for developing conservation planning and establishing ecological compensation mechanisms in water conservation areas. In this paper, based on the land use, meteorological, soil, and field observation data of the research area, the InVEST water yield model is used to estimate the water supply of the ecosystem in the Zhangcheng District. The model quantitatively analyzes the spatiotemporal distribution characteristics of water supply services in the basin and the influence of different topographic factors. The results show that the average supply of ecosystem water in the Zhangcheng District is approximately 45 mm, and there is a degree of spatial heterogeneity. The total water supply in the Zhangcheng District is relatively small. The water resource supply in the southwest is relatively small, the rainfall in mountainous forest areas in the southeast is high, its water supply is higher, and the supply of forest land water is relatively high. The high-value areas are mainly distributed at 1500 to 3500 m and 15°~40°; the water supply on the sunny slope is greater than that on the shady slope. With the increase in altitude and slope, the water supply in the basin tends to increase first and then decrease.

**Keywords:** InVEST model; water supply function; ecosystem services; Zhangcheng District

## 1. Introduction

Forest water conservation has become one of the most crucial issues in forest ecosystems [1]. Moreover, the forest canopy, detritus layer, and soil layer are regularly recognized as the main redistribution areas [2,3] that effectively conserve soil moisture and regulate river flow [4,5]. Water resource supply is important in maintaining regional biodiversity and other key ecosystem functions [6,7], affecting the region's population, socio-economic development, and layout [8]. With global climate change, surface water shortages, and the rapid deterioration of the water environment [9,10], quantitative spatial and visual assessment of the water supply capability of regional ecosystems is one of the most important research topics in hydrology and ecology [11–13].

With the development of remote sensing GIS technology and hydrological models [14], more scholars have tried to quantitatively visualize, accurately analyze, and evaluate the function of regional ecosystems through model simulation methods. Simulation methods include the MIKE System Hydrological European (MIKE SHE) model [15], Temperature Vegetation Dryness Index model (TVDI) [16], Soil and Water Assessment Tool (SWAT) model [17], Soil Conservation Service Curve Number method (SCS-CN) model [18], and Integrate Valuation of Ecosystem Services and Tradeoffs (InVEST) model [19]. The InVEST model quantitatively estimates the water supply of different landscapes based on the water balance principle, considering the spatial differences in soil permeability and topography

under different land uses [13,20–22]. The parameters and characteristic data requirements of the model are low [23], and it can use empirical parameters and quantify ecosystem service functions in the form of thematic maps [24].

Thus far, the InVEST model has been successfully applied by scholars in the assessment of regional ecosystem services on the north coast of O'ahu Island, Hawaii [25], the San Pedro River basin in Arizona [26], the Beijing mountains [4], the upper reaches of the Han River [27], Sanjiangyuan [28], and other regional ecosystem services. However, due to the large geographical and environmental differences between regions, it is crucial to adjust the model parameters according to each research area's characteristics, especially in large- and medium-scale areas where comprehensive field monitoring data are lacking [29]. Simultaneously, significant spatial and temporal differences in the water supply capacity of different ecosystems are closely related to factors such as precipitation, evaporation, land use/overlay changes, topography, soil permeability, and vegetation transpiration [22]. For example, different vegetation types (or ecosystems) have different effects on hydrological factors [30], such as rainfall interception, seepage, evaporation, the distribution and growth status of vegetation, and topographic factors (such as slope direction and altitude), which indirectly or directly affect the spatial pattern of the water supply [31]. Therefore, this paper analyzes the spatial and temporal differentiation characteristics in different ecosystems and clarifies the relationship between water supply and meteorological factors, topographic factors, and land use. Moreover, to provide an important reference for ecological environment construction and protection in the Zhangcheng District, the key driving factors affecting changes in the water supply are examined.

The cities of Zhangjiakou and Chengde (referred to as the Zhangcheng District) are located in the northern part of Beijing and are key in protecting Beijing's water sources, preserving sand sources, and improving the ecological environment. However, due to special location characteristics and historical reasons, the economic development of the Zhangcheng District is relatively slow, regional ecological poverty is prominent, and ecosystem services such as water supply function in river basins are changing, triggering new ecological and environmental problems. Therefore, it is crucial to quantify the water supply service function and the spatial and temporal distribution characteristics of watershed water supply services in the Zhangcheng District. This will clarify the different characteristics of water supply services in different ecosystems and different topographies to provide quantitative visual evaluation results and a scientific basis for key functional zoning, ecological restoration, and ecological compensation of water supply in river basins.

## 2. Research Method and Data Processing

### 2.1. Research Site

The Zhangcheng District is located in northern Hebei Province (113°50′~119°15′ E, 39°30′~42°40′ N). The topography of this area mainly consists of the dam plateau area, the northern Hebei mountain area, and the low hills in northwest Hebei, with the overall terrain gradually decreasing from northwest to southeast (Figure 1a). It belongs to the temperate continental monsoon climate and is characterized by four distinct seasons. The average temperature, precipitation, and potential evaporation in the upper plateau area of the dam are −1~2 °C, 300~400 mm, and 1400 mm, respectively. Due to the influence of topography and latitude, the temperature in the mountains near the dam gradually increases from north to south, and the average temperature and precipitation are 5~9 °C and 400~600 mm, respectively. Zhangjiakou City contains five major water systems: the Yongding River, Chaobai River, Daqing River, Taiqing River, and Inland River. Chengde City employs the province's three northern rivers (tidal white river, white river, thistle canal), Liaohe, and Daling River as water systems. The total water resources in the Zhangcheng District amount to $5.497 \times 10^8$ m$^3$, of which $4.792 \times 10^8$ m$^3$ are surface water resources, $2.873 \times 10^8$ m$^3$ are groundwater resources, and the double-calculated water volume is $2.169 \times 10^8$ m$^3$. In 2015, the forest area in the Zhangcheng District was $4.6607 \times 10^6$ hm$^2$. Forest resources are rich and diverse and have evident horizontal and vertical distribution rules. There

are forests, grasslands, farmland, water, and other ecosystems in the area. Forest and grassland ecosystems mainly include *Picea asperata* Mast, Quercus, Betula, *Larix principis-rupprechtii* Mayr, *Pinus sylvestris*, and *Pinus tabulaeformis* Carr, which play a crucial role in water conservation.

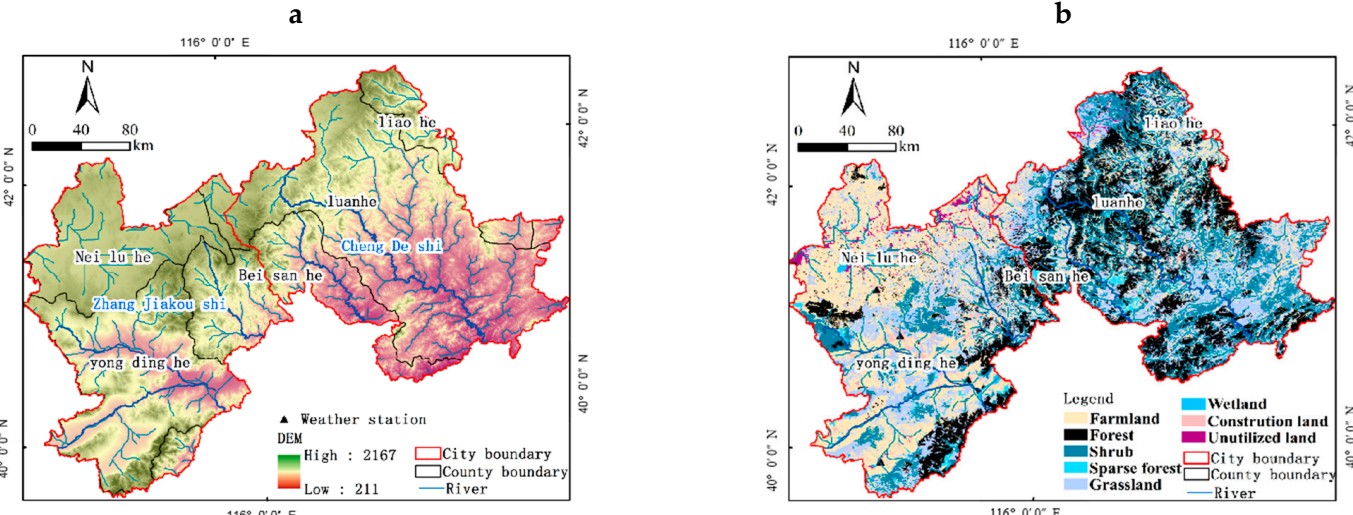

**Figure 1.** The location (**a**) and land use type (**b**) of the study area.

## 2.2. InVEST Water Supply Service Model Method

The InVEST model of water supply service (also known as a water production module) is a raster-based water balance estimation module [32,33]. Moreover, the relationship between climate, terrain factors, and water circulation simplifies the converging process. Assuming that the grid water service reaches the outlet in any of the above ways, the amount of water supplied to each grid cell (including surface production flow, soil water content, dry matter holding capacity, and cover flow) equals the sedimentation. By subtracting the actual evaporation and simulating the spatial distribution of the regional water supply [13,34–36], the specific calculation formula is as follows:

$$Y_{jx} = \left(1 - \frac{AET_{xj}}{P_x}\right) \times P_x \tag{1}$$

$$\frac{AET_{xj}}{P_x} = \frac{1 + w_x R_x}{1 + w_x R_x + \frac{1}{R_{xj}}} \tag{2}$$

$$w_x = \frac{Z \cdot AWC_x}{P_x} \tag{3}$$

$$R_x = \frac{K_{xj} \cdot ET_0}{P_x} \tag{4}$$

$$AWC_x = \min(MSD_x, RD_x) \times PAWC_x \tag{5}$$

Type $Y_{jx}$ is a landscape type $j$ with upper grid unit $x$ annual water supply service (m$^3$); $AET_{xj}$ is the type $j$ upper grid unit $x$ annual average evaporation emission; $P_x$ is the average annual rainfall of the grid unit $x$; $Z$ is the Zhang coefficient, also known as the seasonal factor, which is a constant ranging from 1 to 10 that characterizes the average precipitation characteristics of multiple years determined by the seasonal time distribution and rainfall. For areas where rainfall is concentrated during winter, the Z-value tends towards 10, and for areas where rainfall is distributed throughout the year or in summer, the Z-value tends towards 1; $w_x$ is a non-physical parameter that characterizes the natural climate–soil properties and is dimensionless; R$xj$ is the drying index of the grid element

$x$ on landscape type $j$ and is dimensionless [37]; $AWC_x$ is the water content available to plants; $ET_0$ is the potential dispersion (mm); $K_{xj}$ is the crop coefficient, which is the ratio of the ET of landscape type $j$ on the grid element $x$ to potential dispersion $ET_0$, which is also called the vegetation dispersion coefficient in the model; $MSD_x$ is the maximum soil depth; $RD_x$ is the root depth; $PAWC_x$ is the available water available through indirect calculation of soil texture [38].

### 2.3. Data Processing

InVEST water supply service model input parameters include rainfall, potential evaporation, land use/overburden type, soil thickness, effective soil moisture content, and a biophysical parameter table.

1.  Annual rainfall (P) was calculated based on the rainfall data of meteorological stations in and around the Zhangcheng District (18 stations) from 1990 to 2020, and the data from the National Meteorological Science Data Center (http://data.cma.cn/ accessed on 12 August 2021). Using the Kriging method to interpolate rainfall data, the spatial distribution raster data of the average rainfall in the research area for multiple years were obtained (Figure 2a).
2.  Potential evaporation ($ET_0$). The improved Hargreaves formula, one of the simplest empirical formulas for calculating potential steam emission E, was adopted. Using this formula is more reliable when there are more uncertainties than the Penman–Monteith formula [39–43].

$$ET_0 = 0.003 \times 0.408 \times RA \times (T_{av} + 17) \times (TD - 0.0123P)^{0.76}$$

In the improved Hargreaves formula, $T_{av}$ (in °C) is the average maximum and lowest temperature per day, TD is the difference between the two temperatures, RA is the astronomical radiation (unit : $MJm^{-2}d^{-2-1}$), P represents precipitation (unit : mm/month). Temperature and precipitation data were obtained from the National Meteorological Science Data Center (http://data.cma.cn/ accessed on 12 August 2021); RA numbers were obtained by the regression line method of correction (Figure 2b).

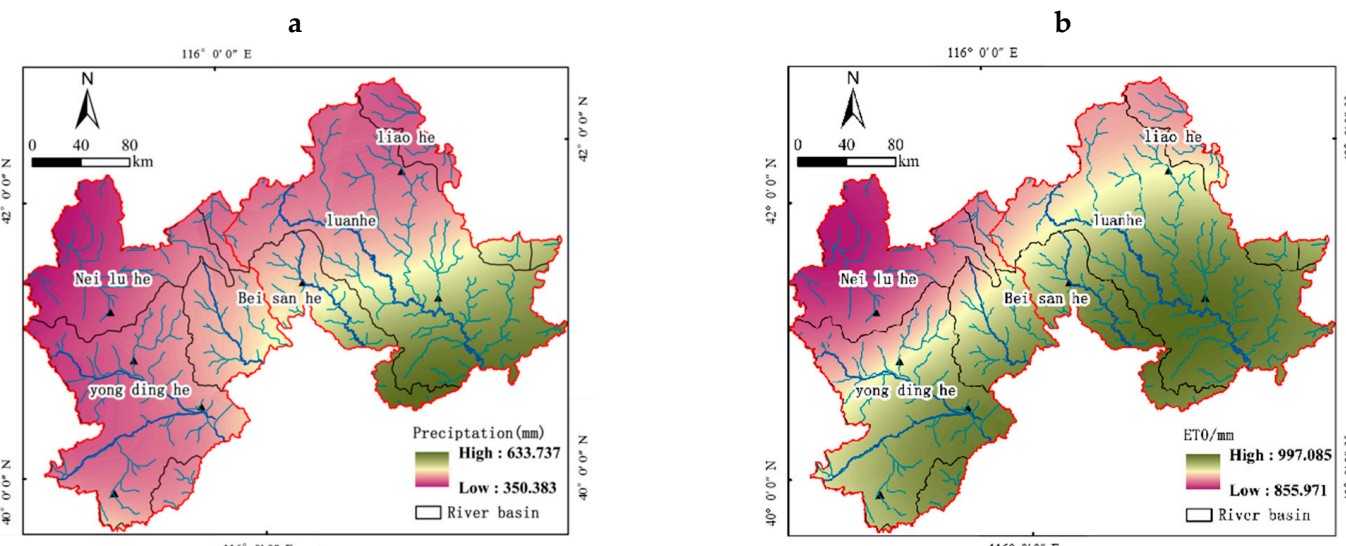

**Figure 2.** Precipitation (**a**) and $ET_0$ (**b**) spatial distributions in the Zhangcheng District.

3.  Land use type (LUCC) was obtained through the Geographic Data Sharing Infrastructure of the Resource and Environment Science and Data Center (http://www.resdc.cn/, accessed on 12 August 2021). It mainly uses Land-sat remote sensing image data as the primary information source and the land use/land cover remote sensing monitoring database (CNLUCC) established through visual interpretation,

combined with the actual landscape types of the research area. Here, the land use type was divided into 12 subcategories in six categories, namely cultivated land, forest land (forest land, shrubland, sparse forest land, other woodlands), grassland (high-covered grassland, medium-cover grassland, low-cover grassland), water (river, lake, and reservoir), construction land, and unused land (sandland, bare land, and high Mountain snow-naked rock) (Figure 1b).

4.  Water content available to plants (%). The available water content of plants was calculated using the method proposed by Gupta [38]. The calculation formula is as follows:

$$PAWC = 54.509 - 0.132sand - 0.003sand^2 - 0.055silt - 0.006silt^2$$
$$-0.738clay + 0.007clay^2 - 2.688OM + 0.501OM^2$$

In the formula, sand, silt, and clay represent the content of sand, powder, and clay in the soil (%), and OM is the organic matter of the soil (%). Soil data (HWSD V1.2) only provide soil organic carbon content. Therefore, to obtain soil organic matter content, the biommelem coefficient was used to convert soil organic carbon content to soil organic matter content. The above method was used to obtain the water available to plants in the Zhangcheng District (Figure 3).

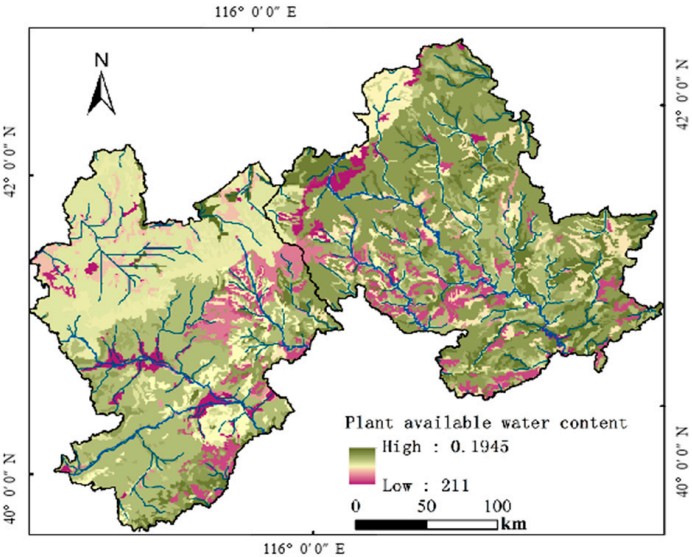

**Figure 3.** Plant available water content in the Zhangcheng District.

5.  Biophysical parameters table. The biophysical parameters table reflects the properties of land use and land cover types in the study area, including land use coding, maximum root depth of vegetation, and evaporation coefficient. The maximum root depth of vegetation data and the dispersion coefficient were obtained from previous research results [41]. Food and Agriculture Organization of the United Nations (FAO) reference values for evapotranspiration coefficients (crop coefficients) (http://www.FAO.org/docrep/x0490E/x0490e00.htm/, accessed on 12 August 2021) and InVEST model database data were generated from dbf data by land use type (landscape type).

## 3. Results

### 3.1. Model Parameter Calibration

It is difficult to accurately verify the prediction results of the model water supply module of InVEST. To verify the accuracy of the simulation results, this study uses Hebei Water Statistics Bulletin data from 1995 to 2015 to determine and validate the relevant parameters of the model; it was used for model parameter calibration in 1995, 2005, and

2015, and for model validation in 2010 and 2015. The Zhang coefficient is based on the precipitation–runoff relationship to obtain the average natural runoff. It is estimated according to the principle that numerically connects the natural runoff. When the Zhang coefficient is 6.8, the model simulation depth is the closest to the actual water production depth, and the relative error remains within 20%, indicating that the model parameter calibration is preferable.

*3.2. Model Verification*

This study used the meteorological and land use data of five periods between 1995 and 2015 as input for simulation calculation and compared them with the obtained actual data to verify that the expected model performance was less than 20% (Table 1). The comprehensive verification results show that in 20 years, the total simulated water production volume and the actual water production in Zhangjiakou City and Chengde City in 2015 are 7.73% and 4.83%, respectively, with an error of 3.73% for Chengde City and 0% in Zhangjiakou City. Thus, the InVEST model has good simulation accuracy, the parameter effect is more robust, and it can estimate the water production volume in the Zhangcheng District.

**Table 1.** Annual water yield model result verification of the InVEST model.

| Model Accuracy Verification | Year | City | Simulated Water Depth (mm) | Actual Water Depth (mm) | Relative Error (%) |
|---|---|---|---|---|---|
| Corrected parameter | 1995 | Zhangjiakou | 42.63 | 36.76 | 15.97 |
| | | Chengde | 42.96 | 42.29 | 12.20 |
| | 2000 | Zhangjiakou | 44.14 | 37.32 | 18.27 |
| | | Chengde | 46.04 | 44.69 | 13.18 |
| | 2005 | Zhangjiakou | 43.57 | 37.88 | 15.01 |
| | | Chengde | 44.76 | 40.64 | 10.14 |
| Verified result | 2010 | Zhangjiakou | 43.10 | 46.00 | 6.30 |
| | | Chengde | 45.48 | 47.20 | 3.64 |
| | 2015 | Zhangjiakou | 42.20 | 42.20 | 0.00 |
| | | Chengde | 44.80 | 38.90 | 3.73 |

*3.3. Characteristics of Spatial–Temporal Distribution of Water Supply Services in Zhangcheng District*

Figure 4 shows that the spatial distribution pattern of water supply services in the Zhangcheng District has minor changes and generally shows regularity. The high-value water supply service zones are concentrated in Huai'an County, Guyuan County, Zhangjiakou City, Xinglong County, and Pingquan County, southern Chengde City. The average water supply service is between 60 and 85 mm; on both sides of the Yanshan-Tahang Mountains, the average water supply service is between 31 and 59 mm. Located in Yanshan-Taihang Mountains, Weichang County, Longhua County, and Zhuolu County, the average water supply service is relatively small, with 20~30 mm. This distribution pattern is related directly to the average annual precipitation and vegetation distribution in the Zhangcheng District. In other words, the regions with high annual precipitation and low vegetation evaporation have a strong water supply capacity. In terms of time, the average water production during the study tended to increase and then decrease; the total water production in Zhangjiakou and Chengde cities increased from $15.60 \times 10^8$ m$^3$ and $16.84 \times 10^8$ m$^3$ in 1995a to $15.94 \times 10^8$ m$^3$ and $17.55 \times 10^8$ m$^3$ in 2005, and then decreased to $15.45 \times 10^8$ m$^3$ and $17.60 \times 10^8$ m$^3$ in 2015.

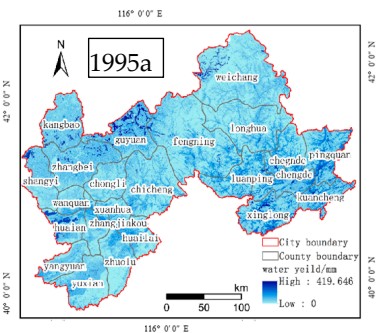
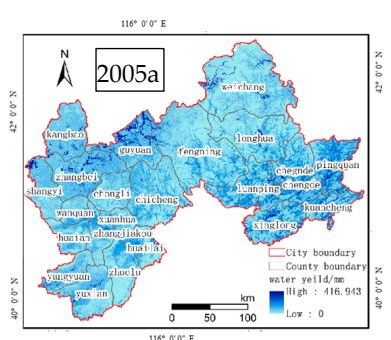
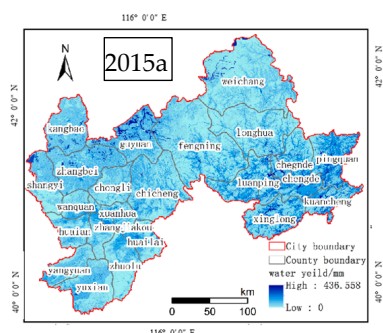

**Figure 4.** Spatial–temporal distribution of water supply service in the Zhangcheng.

*3.4. Distribution Characteristics of Water Yield in the Zhangcheng District by River Basin*

In 2014, General Secretary Xi repeatedly mentioned that during the coordinated development of Beijing, Tianjin, and Hebei, in order to ensure national water security, it is necessary to strengthen the development of the Zhangcheng water conservation area and comprehensively repair and protect the water ecological environment in the Zhangcheng District. The vast majority of the Zhangcheng District belongs to the Haihe River Basin and has evident watershed connectivity. The average water production in different watersheds in the Zhangcheng District varies with time. Spatially, the average water production is manifested as inland river > Liaohe > capital bank > Yongding River > Daqing River (Figure 5a). The total water production and the average water production are similar in time, and this is spatially reflected in the capital management industry > Yongding River > Inland River > North Three Rivers > Liaohe River > Daqing River Basin (Figure 5b).

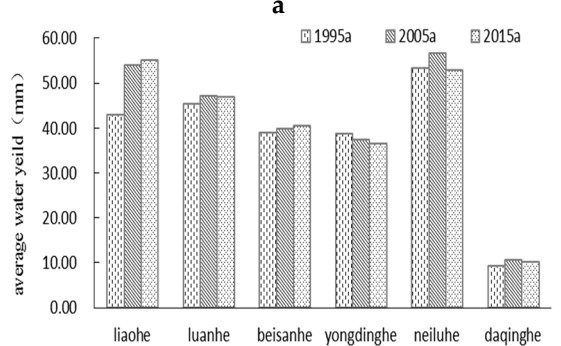
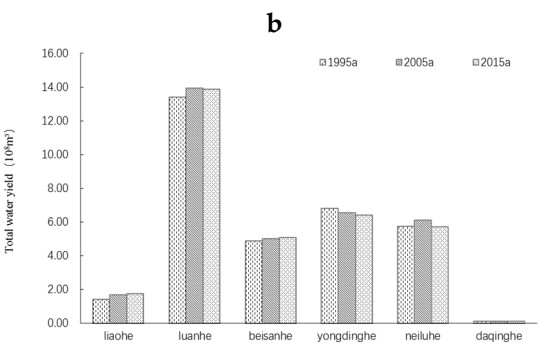

**Figure 5.** Average (**a**) and Total annual (**b**) water supply in different river basins in the Zhangcheng District.

*3.5. Water Yield Distribution Characteristics of Main Vegetation Types in the Zhangcheng District*

The vegetation classification map of the research area was obtained from the Resource and Environmental Science and Data Center of the Chinese Academy of Sciences (https://www.resdc.cn/1995,2005,2015, accessed on 12 August 2021). ArcGIS spatial analysis was used to carry out zoning statistics on the water supply and service capabilities of various vegetation landscape types. The research areas were obtained in 1995. Twenty average (Figure 6a) and total water supply (Figure 6b) were obtained in the third phase of the three vegetation types in 2015. In 1995, the average water supply of each vegetation landscape was in the order: meadow > coniferous forest > other vegetation type > cultivated vegetation > grassland > shrub > grass > broadleaf forest. In 2005, the average water supply of vegetation landscapes was roughly similar to that of 1995. In 2015, coniferous forests had the largest average water supply across the vegetated landscape, followed by grassland and other vegetation types and cropland vegetation, while broadleaf forests remained the lowest. From the perspective of the total water supply, the cultivated landscape plays a leading role. The overall performance is as follows: the farmland ecosystems have the largest total water production, accounting for more than 49% of the total water supply; the

thicket and grasslands account for around 30%, and coniferous forests and other vegetation types' contributions are relatively small (Figure 6b). In the past 20 years, the coniferous forest, broadleaf tree forest, and meadow water supply services have shown a trend of growth followed by a decrease, with shrubs and grasslands showing an annual growth trend; the grasslands' water source supply has declined annually.

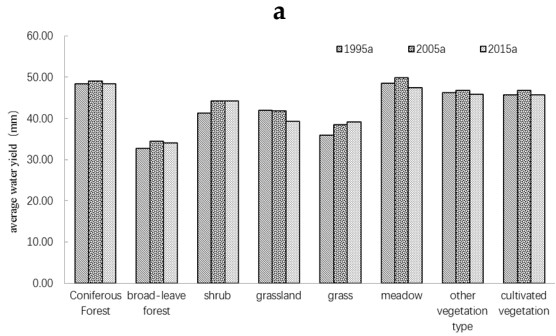 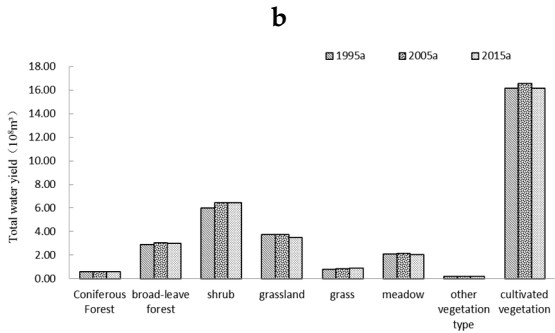

**Figure 6.** Twenty average (**a**) and total annual (**b**) water supply in different vegetation type areas in the Zhangcheng District.

### 3.6. The Influence of Topographic Factors on Water Yield in Zhangcheng District

Topographic factors affecting water supply services in the Zhangcheng District were analyzed from three aspects: altitude, slope, and slope direction. First of all, according to the actual situation of the study area with a large altitude difference, steep peaks and high slopes, and rolling mountains, the elevation of the basin was divided into seven levels: <300, 300~1500, 500~1000, 1000~1500, 1500~2000, 2000~2500, and ≥2500 m. Research shows that the average water production gradually decreases with altitude (Figure 7a); the total amount of water resources is spaced, and the total water supply in the river basin increases with altitude; in areas above 1500 m, the water supply begins to rise with altitude. The regional water supply is primarily concentrated in the altitude range of 500–1500 m, accounting for more than 70% of the total water supply, while the total water supply in areas below 300 m and above 1500 m is relatively small (Figure 7b). Between 1995 and 2015, the total amount of water resources generally showed a rising trend followed by a decline, but the overall change was relatively small.

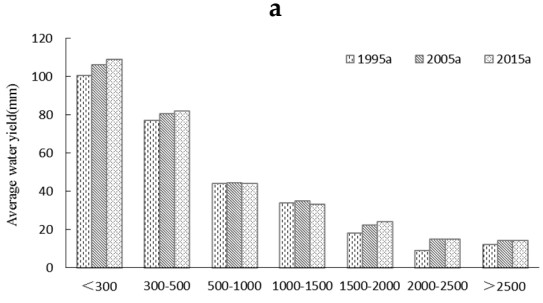 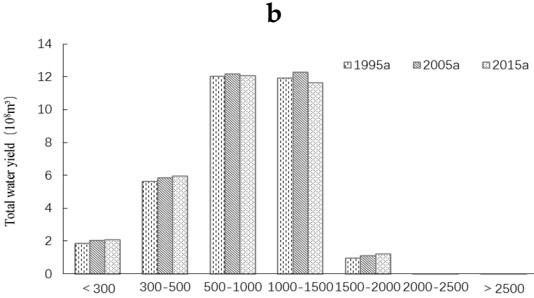

**Figure 7.** The distribution of average (**a**) and total (**b**) water supply service at different elevations in the Zhangcheng District.

DEM was used to extract and grade the slopes in the research area and statistically analyze the water supply distribution in the basin at different slopes (Figure 8). Between 1995 and 2015, spatially, the average water production in the Zhangcheng District increased gradually with the increase in slopes. In terms of time, the changing trend is consistent within different slopes, showing an increasing and decreasing trend. The Zhangcheng District's total water supply is mainly concentrated in the <15° slopes, accounting for more than 49% of the water supply in the study area. In the slope range of <5°, the water supply in the basin is relatively small. The main reason for this situation is closely related to the grid data of different slopes.

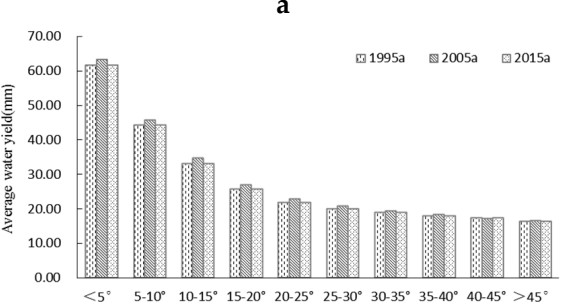
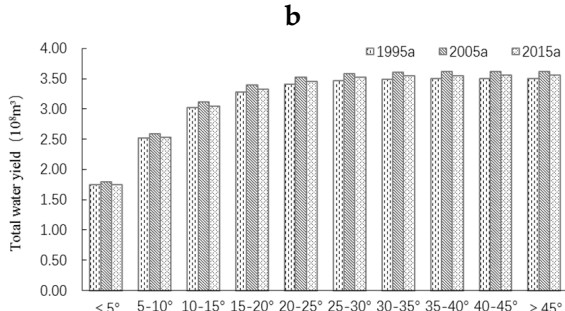

**Figure 8.** The distribution of average (**a**) and total (**b**) water supply service at different gradients of slopes in the Zhangcheng District.

Figure 9 shows that the water output slopes of the Zhangcheng District are slightly different, but the overall water supply of the sunny slopes is slightly larger than that of the shady slopes. The average water supply volume of shady and sunny slopes is 43.35 and 43.63 mm, respectively, and the total water supply in the basin can be expressed as shady slope > sunny slope > half sunny slope > half shady slope. In terms of time, between 1995 and 2015, slopes generally showed a trend of slight growth before decreasing.

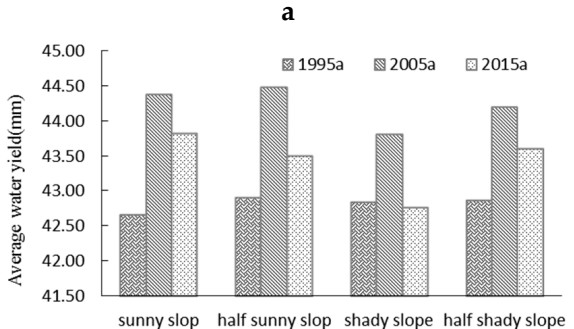
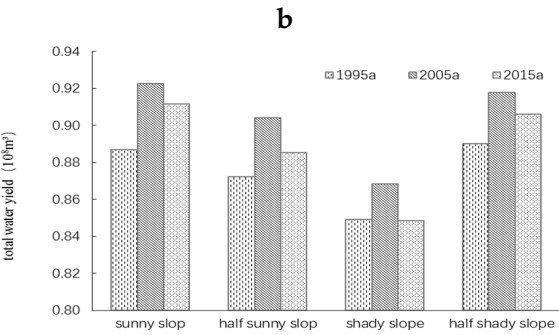

**Figure 9.** The distribution of average (**a**) and total (**b**) water supply service for different slope aspects in the Zhangcheng District.

### 3.7. Analysis of Drivers of Change in Water Production

The impact of climate change on water production in the Zhangcheng District draws a correlation between the depth of water production, precipitation, and evaporation (Figure 10). Over the years, the average depth of water production has been positively correlated with the average precipitation. The potential evaporation emission is negatively correlated, consistent with Li Yiying, Yang Ying, Li Yingying, Wang Culting, and Han Dongxue, and contrary to Gong Shihan. The research conclusions of a positive correlation between water source conservation and evaporation emission are biased. The changing trend of average water production depth and precipitation in this study is the same. The years in which the extremes occur correspond to precipitation extremes, but there are significant deviations from the potential evapotranspiration emission trends, indicating that precipitation is the main factor influencing water production.

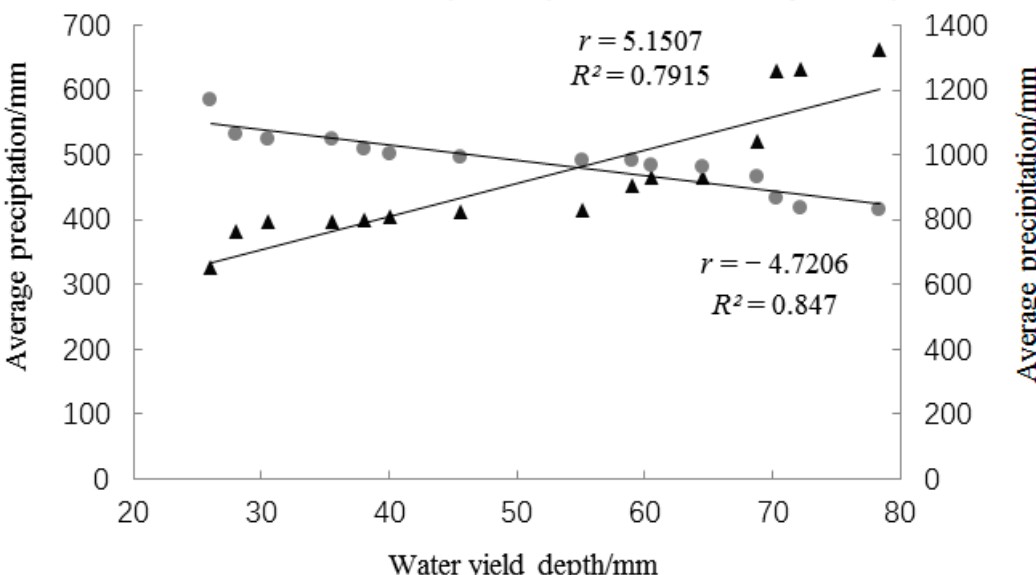

**Figure 10.** Correlations of annual average water yield with precipitation and potential evapotranspiration in the Zhangcheng District.

### 3.8. The Influence of Topographic Factors on Water Production in the Zhangcheng District

A correlation between water production depth and different altitudes and slopes was established (Figure 11). In the range of 0–1200 m, the average water production depth is significantly negatively correlated with altitude ($r = -0.0903$, $R^2 = 0.9528$); in the range of >1300 m, there is a negative correlation with altitude ($r = -0.0173$, $R^2 = 0.8262$), but the trend is relatively flat. Surface runoff has a cumulative effect in the area, which has great potential for runoff formation, so the water production volume change is negatively related to altitude. The correlation between slope and water yield change is the same as that of elevation. In a range lower than 25°, the average water production is significantly negatively correlated with the slope ($r = -9.8983$, $R^2 = 0.9352$); in a range greater than 25°, the average water production negatively correlates with the slope ($r = -0.9552$, $R^2 = 0.9932$), and the trend is relatively smooth.

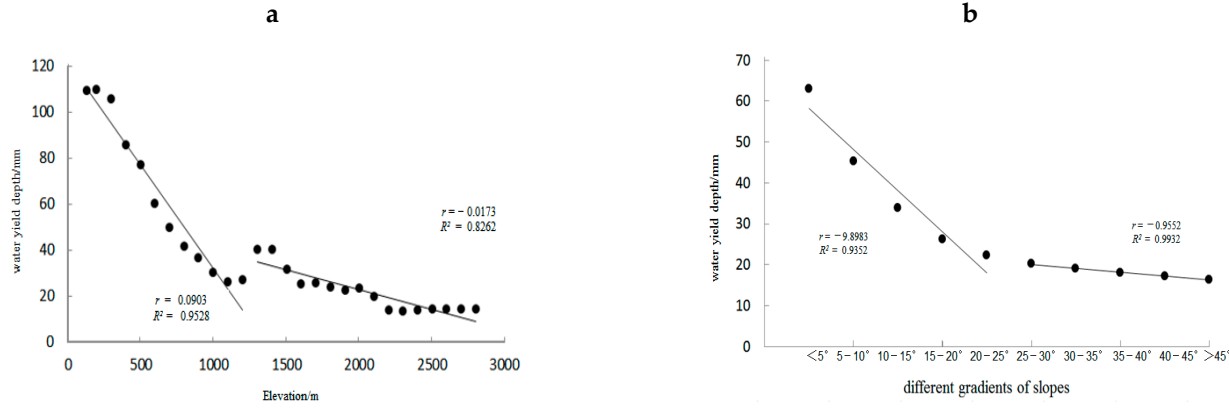

**Figure 11.** Correlations of annual average water yield with different elevations (**a**) and gradients of slopes (**b**) in the Zhangcheng District.

### 3.9. The Impact of Land Use Change on Water Production in Zhangcheng District

Land use is the most critical data source of the InVEST water production model and one of the key factors affecting water production. Table 2 shows noticeable differences in yearly water production between 1995 and 2015, which may be caused by different regional geographical differences, consistent with the results of [35,43], such as in the Zhangjiakou-Schengde region [38] and Shiyang River basin. Statistics on the various land use changes caused over the years, their contribution to water production and conservation, and water production and conservation changes are presented in Table 2. The overall land use change from 1995 to 2015 is as follows: construction land, sparse forest land, grassland, shrubs, woodland, cultivated land. As a result, the water production in construction land and cultivated land has been reduced. Although the area is large, due to the low conservation capacity of construction land and unused land water sources, the contribution to water production is still small. The main reason for the decrease in the overall area of grassland and the increase in water production may be the overall increase in water production due to the increase in regional precipitation over the years. Moreover, the dynamic change in unused land in Zhangjiakou over the years—that is, the unused land area in areas with less precipitation, such as Kangbao and Guyuan—and the decrease in Huailai might influence this decrease. Unused land increases in areas with more water, so areas with more precipitation produce more water than areas with less precipitation. In terms of water production, the most productive areas are cultivated land and grassland, followed by shrubs, construction land, and woodland. Due to the small amount of precipitation in the Zhangjiakou area, the evaporation and emission of forest land are too large, thus reducing water production.

**Table 2.** Changes in water yield caused by various land uses in Zhangcheng District over several years.

| Type | Water Yield Contribution (%) | | | Variation (%) | |
|---|---|---|---|---|---|
| | 1995a | 2005a | 2015a | Area/km$^2$ | Water Yield/($\times 10^8$ m$^3$) |
| Farmland | 50.84 | 54.36 | 51.38 | −7.48 | −2.99 |
| Forest | 4.79 | 2.82 | 2.78 | 10.02 | 40.84 |
| Shrub | 9.26 | 4.98 | 4.13 | 11.94 | 54.55 |
| Sparse forest | 2.50 | 0.62 | 0.53 | 73.50 | 78.55 |
| Grassland | 23.03 | 23.20 | 21.90 | −23.37 | 3.07 |
| Construction land | 5.64 | 5.17 | 12.29 | −105.70 | −122.00 |
| Unutilized land | 3.94 | 8.84 | 7.00 | −73.31 | −81.04 |

### 4. Discussion

1. Between 1995 and 2015, the water supply service in the Zhangcheng District showed a trend of first increasing and then slightly decreasing, and the distribution pattern showed minor changes. This phenomenon is related to watershed meteorological factors (such as precipitation and potential evaporation) and the distribution area of the vegetation distribution pattern from a vegetation landscape type. Precipitation in the watershed of the Zhangcheng District is the primary source of ecosystem water circulation, and potential evaporation indirectly reflects the water consumption capacity of regional ecosystems [35,42]; the bottom surface state and its spatial distribution also affect the distribution pattern of the water supply capacity of watershed ecosystems [44]. Regarding spatial distribution, areas with a high water supply in the Zhangcheng District are mainly distributed in mountainous forest areas with abundant rainfall and lush forest growth, such as Fengning Manchu Autonomous Region, Longhua County, and Waichang County, located in the Yanshan-Taihang Mountains. Although these areas have relatively high rainfall, the air is relatively humid. Coniferous forests have low evapotranspiration and high water production. When old coniferous forests are cut, planted forests and secondary forests do not proliferate, and the water production volume in river basins increases. However,

over time, vegetation growth and restoration entered a mixed growth stage involving shrubs, secondary broadleaf forests, mixed coniferous forests, artificial spruce forests, a forest cover increase, and a water production decline [45–47]. Under the same climatic background, the water supply in the river basin decreased with the restoration of vegetation and the expansion of the proportion of forest land. This phenomenon and its distribution pattern are similar to those in areas such as bridge reservoirs [48], the upper reaches of Miyun reservoirs [49,50], the Lancang River basin [51], or water supply distribution patterns. However, the distribution of the parallel area of the Three Rivers [52] is slightly different, mainly manifested in ice and snow glaciers. This is mainly because the parallel area of the Three Rivers belongs to the high-altitude zone, and the ice and snow cover is extensive, so its water production is high. The Zhangcheng District is relatively unaffected by ice and snow glaciers and has primarily seasonal snow accumulation produced in the basin. The influence of the water process is relatively weak. In addition, the ability of the ecosystem to intercept rainfall is weak, the root system is shallow, and the cultivated area is large, so the total supply of farmland water is relatively large.

2.  Among different topographic factors, the average water production in the Zhangcheng District decreases with the increase of altitude and slope, and the water supply to the shadowed slope is greater than that of the sunny slope. This is similar to the results of other scholars [53] and is mainly influenced by the characteristics of rainfall and topographic differences. The highest precipitation is found at elevations of 500–1500 m. In the windward slope area, the forest is widely distributed, the topographic rain characteristics are apparent, and the potential evaporation is not large. For example, there are scattered trees such as deciduous broad-green broadleaf forests and broadleaf mixed forests, and the forest water source has a strong conservation capacity, so the water supply is large. In Kangbao, Zhangbei, Shangyi, and other places in the plateau area of the dam, in addition to cultivated land, there are also industrial and mining lands, residential land, and other construction lands. Because the total area is minimal, the total water supply is relatively small. At the same time, these areas have crossed the forest line of most forest vegetation, and the surface vegetation landscape has gradually been irrigated with grassland and cold grassland. Alpine sparse vegetation and bare rock are replaced by relatively little rainfall and the weak conservation capacity of vegetation sources, so its water supply is relatively small.

## 5. Conclusions

1.  In the Zhangcheng District ecosystem, the average water supply is approximately 43.5 mm, which shows a certain regularity in space. The high-value areas are mainly distributed in the mountain forest areas, with the most prominent water conservation forests in the dams of Yanshan-Tahang Mountain. Most low-value areas are concentrated in the upper plains or altitudes of dams with low rainfall and frequent human activities. From the perspective of land use type, the water production capacity of the farmland ecosystem in the Zhangcheng District is relatively high. In terms of time, the water supply services in the basin show a trend of initial growth and then a slight decrease.

2.  The overall water supply in the Zhangcheng District tends to rise first and then decrease with the altitude increase. The water supply in the areas with steep slopes is slightly higher than in areas with mild slopes, and the high-value areas appear at 500–1500 m and 300–500°, respectively; the water supply volume of Yin and Yangpo is greater than that of the half-yang slope and half-yin slopes.

3.  The introduction of the InVEST model provides a feasible method for estimating the spatial distribution of water supply services in large- and medium-scale regions. However, there may be uncertainties in the study results due to the simplified model structure and methodology and the lack of large field stations and long-term experimental observation data in the study area. Qualitatively, it is suggested that in future

research, based on the evaluation model and its parameter suitability, the observation, localization, and verification of field data should be further strengthened, focusing on developing China's localized evaluation model to ensure the credibility and reliability of the evaluation results.

**Author Contributions:** Conceptualization, R.L. and B.W.; methodology, R.L.; software, R.L.; validation B.W., X.N. and Q.S.; formal analysis, R.L.; investigation, R.L; resources, Q.S. and R.L.; data curation, Q.S.; writing—original draft preparation, R.L.; writing—review and editing, R.L.; B.W., X.N.; visualization, R.L.; supervision, X.N. and Q.S.; project administration, X.N.; funding acquisition, X.N. All authors have read and agreed to the published version of the manuscript.

**Funding:** This research was funded by the Central Non-profit Research Institution of CAF, grant number CAFYBB2020ZE003 and CAFYBB2020ZD002-2; Ecological positioning observation and ecological function calculation of national public welfare forest construction effectiveness monitoring and evaluation, grant number 2130207-20-201/107.

**Conflicts of Interest:** The authors declare no conflict of interest.

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
