# Peer review of "InVEST Model-Based Spatiotemporal Analysis of Water Supply Services in the Zhangcheng District"

_forests, doi:10.3390/f12081082_

Round 1
Reviewer 1 Report
This manuscript estimated the changes in water yield within the watershed area in China.
In spite of limitation on available data, the authors successfully simulated the water yield and then provided the quantitative results.
However, there are several points that should be improved.
<Major comments>
Abstract just exhibits normal methodology and results without interesting analysis or implications. Authors should emphasize novelty and distinguishing result.
Introduction should be improved and well-arranged, separating the paragraphs.
-Importance of water yield estimation
-Methodology of estimating water yield (and why we use InVEST, rather than using other methods)
-Need of estimating water yield in Zhangcheng District (already exists)
The results seem not significant and distinguishing compared to previous studies, simulating InVEST model.
As this journal has taken attention as one of the prominent forestry journals, novelty is essential.
Only providing a case-study result seems not suitable to this journal.
Discussion should be improved.
Current version seems to repeatedly express the result and not to provide interesting implication and suggestion, derived by the results.
Especially, practical implication of the result to reality (e.g., policy implication) needs to be provided for the readership of this journal.
<Minor comments>
Please refer scientific expressions (especially En-dash).
Spacing between words should be corrected. There are so many errors.
Extensive improvement on English writing seems necessary.
Author Response
First of all, thank you very much for your valuable suggestions.Secondly, because my writing ability is indeed insufficient, it is in urgent need of improvement.For the questions raised by the teacher, I tried my best to modify in the article, of course, there are still some deficiencies, also hope that the teacher continue to put forward valuable opinions, I will continue to modify, in order to improve their writing ability, very grateful!Best wishes!

Reviewer 2 Report
In the submitted manuscript "Spatiotemporal Analysis of Water Supply Service in Zhangcheng area Based on InVEST Model" I miss precisely and clearly defined goals / aims (usually at the end of the introduction). Furthermore, unfortunately, the authors do not deal in the introduction with the significance of the paper and the reference to another similar research. So, I could not evaluate the paper in terms of the novelty. The results of this article are certainly interesting for experts who "move in a given location (Zhangcheng area)", but in general, from a broader (global) point of view, I lack the importance of content and originality. These points and next comments (18) must be resolved before publication.
Comments:
1) p.1, l.20: What is "yeild"? I assume it is "yield". Výraz "yeild" se objevuje i v obrázcích 5, 6 a 10.
2) p.2, l.85: What exactly does "preventing sand sources" mean (how do you explain it)?
3) Very long sentences appear in the text (the reader gets lost), eg lines: 19-23, 89-95 and more.
4) p.3, l.98: "(the city of Zhangjiakou and Chengde)" already precisely defined/explained in line 83.
5) Chapter 2.1 Research site: no references are given (where did the data come from?).
6) p.3, l.113: What does mean "2015a"?
7) Missing reference to Figure 1 in text (no link between text and Fig).
8) p.3, l.120: Labels A and B? (Figures should not be below each other). As well as Fig2.
9) No unified references (lines 161, 167, 173, 179, 189, 207, 269,...)
10) p.5, l.169: What does mean "average average maximum"?
11) p.5, l.182-187: Relatively late description (link between text and Fig) Figure 1b.
12) p.6, l.199: The units for Plant available water content are not shown in Figure 3 (not even in the text).
13) p.7, l.252-259: Where is the context with Figure 5a?
14) p.9, l.305: What is "DEM"?
15) p.9, l.316-321: What is Yang slope, Ying slope, Yinbo, Yangpo, semi-yang slope and others?
16) p.10, l.329-330: The research results should be referred in references.
17) p.11, l.337: It is necessary to distinguish between vertical axes (main and secondary). For example, color as a result (black axis and gray axis).
18) p.11, l.349: Is the value "r" correct? Similarly in Figure 10 (r = 5.15 and -4.72)? Check please.
Author Response

(The authors gave the same response as above.)

Round 2
Reviewer 1 Report
Dear authors,
Thank you for your effort to improve the quality of manuscript.
The manuscript has improved with consideration of my previous comments.
Just, the Discussion needs to be simply supplemented. For example, references should be added (Lines 400-404, 412-418).
I strongly recommend the authors to take an English-Editing service to improve the quality of English expression. Because of this request, I decided Major revision. Please take this service for better readership of this international journal.
Best regards,
Author Response
dear,
Thank you very much for your valuable comments on my article. I am struggling to modify the sentences these days. Please forgive me for any delay.I'm sorry!
